# Interpretation and Working through Contemptuous Facial Micro-Expressions Benefits the Patient-Therapist Relationship

**DOI:** 10.3390/ijerph16244901

**Published:** 2019-12-04

**Authors:** Felicitas Datz, Guoruey Wong, Henriette Löffler-Stastka

**Affiliations:** 1Department of Psychoanalysis and Psychotherapy, Medical University of Vienna, A-1090 Vienna, Austria; felicitas.datz@meduniwien.ac.at; 2Faculté de Médecine, Université de Montréal, Montréal, QC H3T 1J4, Canada; guoruey@gmail.com; 3Teaching Center, Medical University of Vienna, A-1090 Vienna, Austria

**Keywords:** affective interplay, working alliance, doctor-patient-relationship, micro-expressions, interpretation and confrontation

## Abstract

Introduction: The significance of psychotherapeutic micro-processes, such as nonverbal facial expressions and relationship quality, is widely known, yet hitherto has not been investigated satisfactorily. In this exploratory study, we aim to examine the occurrence of micro-processes during psychotherapeutic treatment sessions, specifically facial micro-expressions, in order to shed light on their impact on psychotherapeutic interactions and patient-clinician relationships. Methods: In analyzing 22 video recordings of psychiatric interviews in a routine/acute psychiatric care unit of Vienna General Hospital, we were able to investigate clinicians’ and patients’ facial micro-expressions in conjunction with verbal interactions and types. To this end, we employed the Emotion Facial Action Coding System (EmFACS)—assessing the action units and microexpressions—and the Psychodynamic Intervention List (PIL). Also, the Working Alliance Inventory (WAI), assessed after each session by both patients and clinicians, provided information on the subjective quality of the clinician–patient relationship. Results: We found that interpretative/confrontative interventions are associated with displays of contempt from both therapists and patients. Interestingly, displays of contempt also correlated with higher WAI scores. We propose that these seemingly contradictory results may be a consequence of the complexity of affects and the interplay of primary and secondary emotions with intervention type. Conclusion: Interpretation, confrontation, and working through contemptuous microexpressions are major elements to the adequate control major pathoplastic elements. Affect-cognitive interplay is an important mediator in the working alliance.

## 1. Introduction

The importance of the therapeutic relationship to treatment outcome is undeniable. Studies have revealed that “working alliance quality” has a significant impact on the treatment outcome and can even be used as a predictor [1,2,3]. Although many different facets of the therapeutic relationship and its evolution have been described [4], social interaction as an interplay of nonverbal and verbal elements between the therapist and patient to date has not been investigated satisfactorily. Focusing on specific elements of interactions between the clinician and the patient promises interesting insights into the relationship-forming process, perhaps even permitting us to link it to outcome measures and psychotherapeutic success.

Different patient and therapist variables need to be taken into consideration in order to understand a number of complex aspects of the psychotherapeutic relationship. By drawing attention to psychotherapeutic micro-processes, such as countertransference, therapist activity, relationship quality, psychic structure, the quality of object relationship, successful interaction, and “positive” affect regulation and their correlations, a broad field of relationship-generating aspects of psychotherapeutic interaction can be accessed.

Verbal speech, of course, is the therapist’s central mean and tool, with which he or she acts mainly consciously. Nevertheless, the significance of nonverbal communication is also well-known in psychotherapy research [5]. It is therefore remarkable that nonverbal communication is not firmly entrenched in the training curriculum for practitioners. Studies show that 60–80% of all communication (in treatment) is nonverbal [6,7]. A sizable number of nonverbal events are unconscious and can often convey the patients’ emotional and psychological state more incisively than verbal communication can [8].

Moreover, there is a paucity of studies on the correlation between nonverbal and verbal behavior in therapeutic treatment [9,10], even in chronic illnesses such as cancer, they are taken into consideration and studied as communication strategies, in the hopes of improving non-verbal behavior [11]. The strong interplay between verbal and nonverbal communication in therapeutic conversations, as well as their importance, have been discussed on several occasions [12,13,14].

Because of their measurability, several studies focus on facial expressions and their meaning in psychotherapeutic treatment [15,16,17]. Often, subconscious interaction patterns are implemented through nonverbal signals. If the therapist resists this relationship offer verbally as well as nonverbally, the appearance of pathological relationship patterns can be prevented, which may increase the treatment success [18].

Later on, it was shown that facial affective behavior is a good indicator of the relationship balance in successful therapies [19]. However, as a large number of facial expressions are usually detected in any given interpersonal interaction [20], it should not be surprising that many of the visible expressions in conversations of two or more people are ingenuine, being very consciously controlled and displayed. Yet, a major part of mimic is in fact expressed and received unconsciously [19,21,22]. These involuntary expressions need to be revisited in terms of their function.

Overall then, it is absolutely essential in psychotherapy research to investigate facial affective behavior during sessions in order to elucidate the underlying processes at play in the therapeutic interaction. Taking a more in-depth look means examining microprocesses in greater depth. Such an enquiry in psychotherapy may hold important insights into the nature of the psychotherapeutic relationship between therapists and their patients.

In particular, micro-expressions seem an especially promising line of inquiry. So-called micro-expressions, originally discovered in the 1960s by Paul Ekman [23], are defined as facial expressions that last only one-quarter to one-half of a second, and can be understood as either repressed or unconscious expression of emotions [7,24]. Ekman’s research led him eventually to differentiate and name seven basic emotions—*Happiness*, *Surprise*, *Anger*, *Sadness*, *Fear, Disgust*, and *Contempt*.

Accordingly, the occurrence of micro-expressions hint to an unconscious event or process. Dimberg et al. [25] were able to demonstrate that so-called “positive” as well as “negative” emotional reactions can be expressed unconsciously through micro-expressions. Especially when it comes to important aspects of emotional face-to-face communication, the authors found great importance in the unconscious course of actions.

In addition, in some therapeutic encounters recorded on video, the sessions with a high frequency of nonverbal micro-affectivity were rated as being “relevantly changing” [26]. If that is the case, it shows that common analyses between cognition and affect might be abortive, as they do not capture the speechless forms of affective expression. Periods of silence for instance, are often loaded with intense transference [27].

In this light, it seems promising to investigate the coherent occurrence of verbal intervention and unconscious facial micro-expressions, as an expression of unconscious implicit structures in the therapeutic setting [28]. The investigation of the correlation between verbal and nonverbal occurrences as it pertains to the working alliance could lead to a deeper understanding of the effectiveness of specific interventions in therapeutic interactions, as well as provide a novel understanding of unconscious and conscious interactions in psychotherapy, thereby contributing to the conceptual elaboration of therapeutic competences. Both of these points are of the utmost importance in the training of therapists, as well in enhancing psychotherapeutic competence in medical doctors, psychologists, and social workers.

In this exploratory study, we aimed to examine the occurrence of micro-processes during psychotherapeutic treatment sessions, specifically facial micro-expressions, in order to shed light on their impact on psychotherapeutic interactions and patient-clinician relationships.

## 2. Methods

### 2.1. Patients

Adult patients in the Psychiatry Department and/or the Clinic for Psychoanalysis and Psychotherapy of Kaiser-Franz-Josef Hospital in Vienna, who were willing to participate, were included in the introduced study. Only patients of adult age could be included, since the psychiatric department of Kaiser-Franz-Josef Hospital does not admit minors. The study design was open, informing participants about the purpose of the study through an informed consent form, which they were asked to sign. Patients and their treating therapists needed to consent in order to participate. Those who were unwilling or unable to participate (due to psychological issues), as well as patients who, in the opinion of their therapists, could not be asked to contribute, were excluded from the study. The ethics committees of both the Medical University of Vienna and Vienna General Hospital approved this study.

Both male and female patients were surveyed alike. In order to assure patient anonymity, all data was encrypted. Data collection took about two hours for each participant, including the therapy session and the filling out of written questionnaires afterwards. In total, this study included twenty different patients at various stages of their respective therapeutic processes and journeys, interviewed at the Kaiser-Franz-Josef Hospital, all of whom were inpatients or outpatients in the psychiatric department there. For each patient, one session had been recorded, and two out of the 20 patients were recorded during two separate sessions, talking to the same therapist as in the previous interview. Thus, in total, 22 different psychotherapeutic sessions were included and analyzed. The sessions were held between the patient and his/her therapist alone. The mean length of the interviews was thirteen minutes.

Patients had a mean age of 50.25 years (min = 20, max = 75). Gender distribution of patients was balanced, resulting in 10 female and 10 male participants. Eight patients were paired with same-sex psychotherapists and the remaining twelve were paired with opposite-sex psychotherapists. Eight patients talked to a highly experienced psychotherapist and twelve patients were paired with a less-experienced therapist. This is ethically justifiable, as therapists with different backgrounds and working experiences work at Kaiser-Franz-Josef Hospital, which makes this standard of treatment quite typical. Eleven patients had been diagnosed with an affective disorder, seven with a cluster B personality disorder, and two with a somatic symptom disorder. Depression, bipolar disorder, and borderline personality disorder were the most frequently assigned diagnoses. Patients with autism, Asperger syndrome, schizophrenia, schizoid, or schizotypal personality disorder were excluded, given their known difficulty reading facial expression. Seventeen patients were inpatients staying at the clinic, whereas three patients were outpatients of the psychiatry department.

### 2.2. Therapists

Therapists from different institutions with different methodological and clinical backgrounds participated in the study. They assessed current symptoms, functioning, life and work situation, and biographical background. Age was not assessed, due to the small sample size and the need for anonymization, but the experience of the clinicians was taken into account. The correlation yielded no significant results. A distinction was made between therapists with more than 15 years of clinical experience and therapists in training. This factor will help to enable a distinction between more- and less-experienced therapists, differences in their clinical judgement, and the speed with which they develop their relationships with their patients.

Seven therapists—two male, five female—were willing to contribute to the present study. Three of the seven were psychoanalysts, three were therapists without a psychoanalytic background, and the last, being a psychiatrist, had no psychotherapeutic training at all. Three therapists had more than fifteen years of clinical experience and were therefore classified as being “highly experienced”, three therapists were in training which led to them being labeled as “less experienced”, and one therapist taking interviews had not had any prior clinical experience but was planning on starting his training soon thereafter.

### 2.3. Video Recording of Sessions/Rating of Microexpressions

Both therapist and patient were recorded with separate high-definition 4K video cameras recording sixty frames per second, such that each party was filmed from a front view. This was necessary in order to be able to detect the very subtle facial movements forming the micro-expressions. Only five minutes of each recorded session (minutes 5:00–10:00) were analyzed with a split-screen technique in order to reduce data file size and the time required for analysis.

Evaluation of micro-expressions during interviews was conducted in various stages that we have described in detail previously [auto-réf.]. The facial expressions were assessed without content of what was said, by muting the videos. The transcripts were conducted thereafter.

The reliability of the coding was ensured by a training course at the Study Group of Prof. Eva Bänninger-Huber from Innsbruck University, followed by a standardized final test, which was independently evaluated by Paul Ekman’s group. The reliability of the final test was very good (r^2^ > 0.80).

The raters conducting the content analysis and the analysis of FACS and EmFACS as described in [auto-réf], were all blinded regarding the patients’ diagnosis, as well as the therapists’ theoretical background and experience.

### 2.4. Psychodynamic Interventions List (PIL)

The videotaped sessions were transcribed and rated using the Psychodynamic Intervention List [13], with computer assistance from ATLAS.ti. The Psychodynamic Interventions List (PIL) is an instrument to identify and rate psychodynamic verbal techniques. It has shown good inter-rater reliability [29], and consists of 37 categories, associated with three dimensions: Intervention Form (24 categories)Thematic content (9 categories)Temporal focus (4 categories)
The instrument comes with a manual available in long and short versions in both English and German.

These intervention categories were rated by four evaluators using the German manual after the twenty-two videotaped therapy sessions were transcribed via ATLAS.TI. Afterwards, the intervention sessions were grouped into two main categories of techniques, interpretive or supportive, according to Mayring’s technique of qualitative content analysis. Interpretative techniques can be understood as enabling insight into recurring conflicts [30] and/or facilitating the expression of problems in order to encourage and promote insight [31]. Supportive techniques, on the other hand, are defined as those strengthening the abilities of a patient that are hitherto absent or only partially developed, as well as improving his or her level of functioning and ability to adapt to certain situations [13]. The various interventions of each session, rather than the total sessions, were rated as being either supportive or interpretative.

### 2.5. Working Alliance Inventory-Short Revised (WAI-SR)

The Working Alliance Inventory-Short Revised (WAI-SR) is a measure of the therapeutic alliance that assesses three key aspects of the therapeutic alliance—(a) agreement on the tasks of therapy, (b) agreement on the goals of therapy, and (c) development of an affective bond [32]. Each participating patient and their therapist partner evaluated the quality of their working relationship by filling out the WAI-SR. As different social statistic parameters such as age, gender, and primary/secondary socialisation were taken into account with respect to the building of the patient-therapist relationship, these characteristics also needed to be surveyed. Therefore, patients were also asked to fill out an additional questionnaire detailing such demographic characteristics. All these inquiries took place immediately after the interview session.

### 2.6. Statistical Analysis

All statistical analyses were performed using the IBM SPSS Statistics 19 software (IBM Corp., Armonk, NY, USA). For all analyses, the significance threshold was defined as *p* ≤ 0.05. The data about patient sex, age, and occupation were extracted using descriptive statistics. 

The differences between therapists and patients with respect to facial affects analyzed via FACS were assessed with Mann-Whitney’s *U* test. Because of the small sample size, the corresponding effect sizes for these group comparisons were of great interest. The effect sizes were calculated by dividing U by its maximum value, which is the product of the Ns for the two groups. This corresponds to the probability that a person in one group will be higher than a person in the comparison group. The standardized mean group difference, Cohen´s d, and the conventional benchmarks for interpretation (small effect: r ≤ 0.10, medium effect: r ≤ 0.30, and large effect: r ≤ 0.50; [33]) were applied. The differences in the timing of facial affect were calculated using a univariate ANOVA. To assess the relation between intervention and facial affect with respect to the emotion it manifested, the different types of interventions were summarized, resulting in two different groups of interventions (Interpretative and Supportive) and were analyzed using Spearman´s rank order correlation (using two-tailed significance levels). Likewise, the relation between working alliance and facial affect was analyzed. In order to evaluate the differences between all three levels of psychotherapeutic education (psychoanalytic background, no psychoanalytic background, and no therapeutic background), a Kruskal-Wallis and a Mann-Whitney-U test were calculated. Bonferroni correction was applied, so all the effects are reported at a 0.003 level of significance.

## 3. Results

The patients’ affect expressions reflected the psychopathology and diagnoses quite well, as the frequency percentages were quite low. Table 1 gives an overview of the therapists’ affective engagement. Furthermore, as shown in Table 1, scores for the working alliance established between the therapist and patient (as rated by patients) were also quite low, reflective and typical of triage work in an inpatient acute psychiatric ward providing services for a large geographical region, where patients are hospitalized generally only for short periods of time before being referred to more specialized clinics/departments for more targeted treatment, as necessary. Table 1 also shows interventions provided by the therapist: item numbers 1 to 16 were rated and clustered as being interpretative/confrontative types of intervention, whereas numbers 17 to 24 were classified as being more supportive.

### 3.1. Microexpressions vs. Intervention Type

Interestingly, as can be seen in Table 2, in both patients and therapists, contempt was mainly associated with interpretative types of intervention, whereas surprise was associated with more supportive types of intervention. Furthermore, in the group of patients, we found a high association between supportive types of intervention and facial affects of joy and disgust. No significant differences in terms of the frequency of expression of any of the basic emotions were found between male and female therapists or male and female patients, respectively. Also, whether patients or therapists were paired with someone of the same gender or not in the interviews similarly had no significant impact on such frequency of expression.

Additionally, we analysed the timing of facial affect. A univariate ANOVA revealed a significant difference in timing between patients and therapists (F (1189) = 9.091, *p* = 0.003). Facial affect displays were seen earlier in therapists (m = 06:48.64, SD = 01:52.71) than in patients (05:47.66, SD = 02:11.68). The remaining facial affects (anger, sadness, and fear) were left out of the table as there were so few of them (e.g., anger 1, sadness 1, and fear 3 out of 192 action units).

### 3.2. WAI-SR Working Alliance Evaluations

As outlined earlier, the therapeutic working relationship was measured using the *Working Alliance Inventory-Short Revised* (WAI-SR). We analyzed whether supportive and interpretative forms of intervention affected working alliance differently using a Mann-Whitney-U test. Interestingly, as shown in Figure 1, interpretative forms of intervention were associated with higher WAI scores compared to supportive forms (Mann-Whitney-U = 2283.5, *p* = 0.000, d_Cohen_ = 0.67). Likewise, having different psychotherapeutic backgrounds had an impact on working alliance (χ^2^ (2) = 27.545, *p* = 0.000). There was no significant correlation between the clinician’s experience and other measures. However, we found that sessions with psychoanalytically trained clinicians scored a higher WAI. Post hoc analysis revealed that a psychoanalytical background resulted in higher working alliance scores with a mean rank of 121.65, compared to a mean rank of 71.59 for psychotherapists without psychoanalytical education, and a mean rank of 86.93 for those without any kind of psychotherapeutic education. The patients’ WAI-SR scores were not related to their own facial affects; however, facial affective displays of the therapists were associated with patients’ WAI scores—frequent displays of joy in the therapist were associated with lower scores on the working alliance inventory (Table 3).

## 4. Discussion

Within the scope of this exploratory study, the investigated micro-processes were—(a) the clinicians’ and patients’ micro-expressions measured with the emotion facial action coding system; and (b) the therapists’ intervention types, as assessed using the PIL and qualitative content analysis on the twenty-two psychiatric interviews obtained. In addition, these measurements were studied in connection with the WAI, surveyed after each session.

We analyzed whether supportive and interpretative forms of intervention affected working alliance differently, using a Mann-Whitney-U test. Interestingly, interpretative forms of intervention were associated with higher WAI scores compared to supportive ones. Interestingly as well, the different psychotherapeutic backgrounds had an impact on working alliance. *Post hoc* analysis revealed that therapists with a psychoanalytical background had, on average, higher WAI-SR scores compared to those therapists without psychoanalytical education or those without any kind of psychotherapeutic education.

The relation between working alliance and facial affect was analyzed using Spearman´s rank order correlation. The patients’ WAI-SR scores were not associated with their own facial affects; however, therapists’ facial affective displays were associated with patients’ WAI scores—frequent displays of joy in the therapist were associated with lower scores on the working alliance inventory.

Taken together, a certain picture emerges—on the one hand, we found that interpretative/confrontative interventions are associated with displays of contempt in both the therapist and the patient; on the other hand, our results showed that interpretative/confrontative interventions are linked to a higher WAI-SR score. Although patients’ WAI scores were not related to their own facial affects, we suspect a pattern here that we find noteworthy and plan to investigate in greater detail. The results suggest that both a good satisfying working alliance and displays of contempt have one factor in common—interpretative/confrontative intervention.

If this is the case, it seems promising to think about, research, and clarify what the connection might be between a good working alliance and displays of (unconscious) contempt within the therapist-patient interaction, as this seems contradictory at first sight. We propose that this contradiction is only apparent. We believe it to be a result of the complexity of affects and the interplay of primary and secondary emotions. As EmFACS can only detect so-called primary or basic emotions, the instrument fails to yield affects deriving from the displayed primary affect, leaving this task to interpreters of the EmFACS results.

The literature offers several papers [34,35] on the close resemblance of contempt and envy. For Reed [36], contempt is primarily associated with shame and related to envy later on. Miller [37] talks about the child’s contempt for the breast, a very pristine scene of envy, and its important role for the human psyche.

If we can begin to regard contempt and envy not purely as something unpleasant and reprehensible, we may find it less odd that feelings with such negative connotation can in fact be nurturing and facilitating for the therapeutic alliance. Envy, in particular, was one of the many research topics of Melanie Klein [38]. She emphasized its importance and precisely described its development and function in the early mother-child interaction [39].

It is fascinating that something so counterintuitive at first glance should benefit any form of psychotherapeutic treatment. Yet the results of this study provide good reason to believe that it is very essential to facilitate a working alliance in which all forms of affects can take place, including hostile and destructive feelings against oneself or his/her vis-a-vis. If the therapist-patient dyad allows for such an open environment, and therapists do not shy away from giving confrontative and seemingly unpleasant interventions, the patient may be able to drag his or her internal patterns into the treatment sessions, and therefore experience a better working alliance.

This interpretation can be backed up by our findings on the occurrence of joy. Merten and Krause [19] found that “the frequency of positive facial reciprocity between therapists and patients and the therapist’s reaction to the patient’s offers of displayed joy is a predictor of worse therapy outcome”. This may be why we found that frequent displays of joy in the therapist were associated with lower scores on the working alliance inventory; one might say that therapy is not the place to try to cheer someone up.

We decided to use the PIL, as it shows good inter-rater reliability and is derived from a psychodynamic theoretical background. This might appear counterproductive at first sight, since not every therapist possesses such a background, but as proven by Castonguay and Beutler [40], Barber et al. [41], Wampold [42], and others, therapists tend to use and integrate different techniques from different schools of thought without even necessarily being aware of it. As we are only differentiating between therapists and not labeling interventions to be more or less associated with a certain therapeutic school, we are positive that using a psychodynamic-based measurement will not bias what we find out about any given type of intervention employed or the facial affect displays associated with them.

### Limitations

Several limitations of this study have already been discussed. Many of them are a result of the study design as an exploratory study. For instance, one major limitation lies in the sample size of our study, of course, such that it does not allow for the detection of causality.

A second limitation that merits consideration is the fact that the EmFACS can only discern basic, primary emotions; it is up for interpretation what these basic affects bring with them, and what complex and multi-layered dimensions and meanings they might have for the interacting partners, especially when it comes to the relationship dynamics.

Further, this study is limited in terms of answering questions regarding differences in therapeutic settings. There is cause to believe that different settings will have distinct impacts on relationship-building, as there is major variation in the perception of nonverbal communication. The typical case of the patient lying on the therapist’s couch, for example, makes it impossible for the patient to see the therapist’s body language and likewise limits the therapist’s sight of the patient. This and other setups (interventions via the telephone, for example) could have strong effects on the perceptions of the therapist and patient alike, both in terms of their respective opinions on the effectiveness of the interventions and the strength of the working alliance.

## 5. Conclusions 

We believe the findings of this evaluation contribute to the understanding of the therapeutic relationship. Not only because detailed facets of the relationship and its fostering were discussed, but also, they yielded information on the clinical judgement of therapists and its modification during a therapist’s career. These findings will help to establish knowledge on how to form better therapeutic relationships in order to enhance the likelihood of a successful treatment outcome. Nevertheless, further research is essential. In the first instance, we suggest applying this study design to a larger sample size in order to be able to make further statistical statements about the relations and correlations.

If possible, the video material should also be analyzed more broadly, ideally looking at the entire session. Therefore, we consider qualitative interviews with both therapist and patient concerning the affects on display as reasonable. If the daily routine in the facility allows, we suggest applying the interviews immediately after the session. Apart from questionnaires such as the TRQ, the Transference Questionnaire or the AREQ Affect-Perception and Affect-Regulation Questionnaire may also aid in distinguishing the complex feelings displayed between the therapist and patient. Preferably, future research will also allow for the investigation of control groups. This would enable a comparison between clinician-patient interactions and interactions between therapists and people without a psychiatric diagnosis or even non-clinical, everyday interactions.

## Figures and Tables

**Figure 1 ijerph-16-04901-f001:**
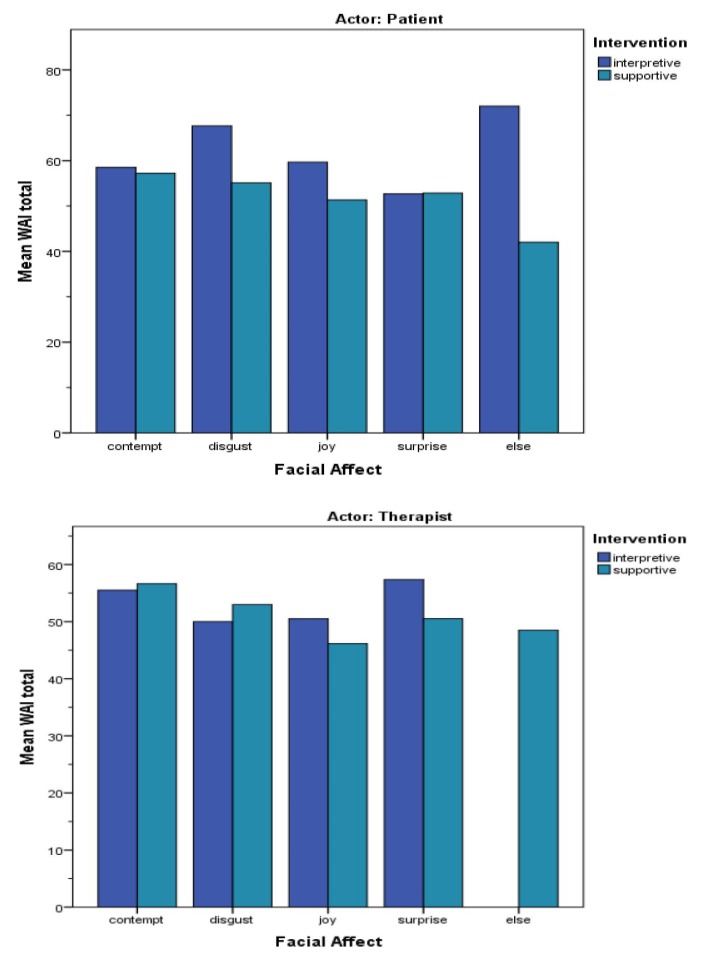
Working alliance inventory score vs patient/therapist basic emotions displayed.

**Table 1 ijerph-16-04901-t001:** Descriptive statistics of patients’ and therapists’ microexpressions, type of intervention, and working alliance.

Emotion of the Patient (FACS/ME)				Emotion of the Therapist (FACS/ME)			
	Frequency	%			Frequency	%	
Contempt	11	20		Contempt	32	23.5	
Disgust	18	32.7		Disgust	12	8.8	
Joy	9	16.4		Joy	19	14	
Suprise	15	27.3		Suprise	70	51.5	
Sadness	1	1.8		Sadness	2	1.5	
Else	1	1.8		Else	1	0.7	
Total	55	100		Total	136	100	
**Descriptives statistics of patient variable: working alliance (WAI)**				**Descriptives statistics of therapist variable: intervention (PIL)**			
		Statistic	Std. Error			Frequency	%
WAI total Mean		53.7	0.921	1 Repeating, paraphrasing, summarizing		13	6.8
95% Confidence Interval for Mean	Lower Bound	51.88		2 Drawing attention to a behavioral and/or cognitive pattern		9	4.7
	Upper Bound	55.51		4 Implicity indicating a parallel		1	0.5
5% Trimmed Mean		53.5		10 Referring to the therapeutic relationship		3	1.6
Median		52		11 Exploring		25	13.1
Variance		161.949		12 Adding new meaning		3	1.6
Std. Deviation		12.726		13 Creating causal links		1	0.5
Minimum		29		14 Interpretation using metaphors		2	1
Maximum		82		15 Encouraging an view or impulse		1	0.5
Range		53		16 Validation		6	3.1
Interquartile Range		11		17 Suggestion		35	18.3
Skewness		0.725	0.176	18 Self-disclosure		4	2.1
Kurtosis		0.372	0.35	19 Association		1	0.5
				20 Expression of mental sympathy		8	4.2
				21 Conveying professional knowledge		15	7.9
				22 Other		48	25.1
				23 Sentence fragments		3	1.6
				24 Single filler words e.g., “hmmm”, “nah”		13	6.8
				Total		191	100

**Table 2 ijerph-16-04901-t002:** Correlation coefficient for facial affect and type of intervention.

Actor	Type of Intervention	Facial Affect			
		Joy	Contempt	Disgust	Surprise
Patients					
	Confrontativ	0.377	**0.662 ****	0.382	0.277
	Supportiv	**0.561 ****	0.229	**0.538 ****	**0.526 ***
Therpists					
	Confrontativ	0.224	**0.454 ***	0.214	0.246
	Supportiv	0.087	0.205	−0.205	**0.486 ***

Note: * *p* < 0.05; ** *p* < 0.01.

**Table 3 ijerph-16-04901-t003:** Correlation coefficient for facial affect and type of intervention.

Actor		Facial Affect			
		Joy	Contempt	Disgust	Surprise
Patients					
	WAI	0.017	0.002	0.256	−0.086
Therpists					
	WAI	**−0.455 ***	0.052	−0.197	−0.289

Note: * *p* < 0.05; ** *p* < 0.01.

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
