# Peer review of "Interpretation and Working through Contemptuous Facial Micro-Expressions Benefits the Patient-Therapist Relationship"

_ijerph, 2019, doi:10.3390/ijerph16244901_

Round 1

Reviewer 1 Report

The authors should be commended for their query of an important topic of non-verbal micro-expressions as it impacts the patient-therapist relationship.  they correctly implicated "working alliance quality" as one of the most important factors predicting treatment outcome and connecting non-verbal communication as important.  They cite Paul Ekman's work and the FACS rubric/technology as important to their work.  I believe when they state the "seven basic emotions" they should preface this to say these are the basic emotions that Ekman has found to have corresponding facial expression.  I have a few questions in the methods

METHODS

Add detail that patients with autism, aspergers, schizophrenia, schizoid, or schizotypal were excluded given their known difficulty reading facial expression I would wonder if the age of the therapist is important was the entire session rated as either "interpretive" or "supportive"?  maybe this needs clarity although the evaluations of micro-expression was described in another paper, it would be valuable to have a brief synopsis here given this is the study topic.  I would specifically add information about how the facial expressions were assessed without the content of what was being said You will also need to display the mean scores for all your measures. Before you calculate effect size you should have the means and confidence intervals

RESULTS

the first table should be a summary of all the scores on FACS, PIL and WAI along with patient and therapist characteristics Wasn't the type of intervention in table 1 "interpretive" and not "confrontive" perhaps you need to show the combined "all therapy types" in another row? where are the other facial affects (anger, sadness, fear) in the table?

The difference in WAI scores between types of therapists seemed striking and perhaps much more significant.  it would be good to comment on this impact of skill level on the type of therapy and working alliance.  

I would be curious about how these findings would change if one were to only look at the FACS of only the experienced therapists.  

Author Response

Comments and Suggestions for Authors

The authors should be commended for their query of an important topic of non-verbal micro-expressions as it impacts the patient-therapist relationship.  they correctly implicated "working alliance quality" as one of the most important factors predicting treatment outcome and connecting non-verbal communication as important.  They cite Paul Ekman's work and the FACS rubric/technology as important to their work.  I believe when they state the "seven basic emotions" they should preface this to say these are the basic emotions that Ekman has found to have corresponding facial expression.  I have a few questions in the methods

METHODS

Reviewer 1

Add detail that patients with autism, aspergers, schizophrenia, schizoid, or schizotypal were excluded given their known difficulty reading facial expression I would wonder if the age of the therapist is important

Many thanks for the helpful suggestions, we included all of them in the manuscript as follows:

We included the sentence concerning schizoid, etc. diagnoses. (now added row 133ff.)

Age was not assessed, due to the small sample size and the need for anonymization, but the experience of the clinicians was taken into account. The correlation yielded no significant results. (now added 144 ff.)

was the entire session rated as either "interpretive" or "supportive"?  maybe this needs clarity although the evaluations of micro-expression was described in another paper, it would be valuable to have a brief synopsis here given this is the study topic.  I would specifically add information about how the facial expressions were assessed without the content of what was being said You will also need to display the mean scores for all your measures. Before you calculate effect size you should have the means and confidence intervals

Not the total sessions were rated supportive or interpretative, but the various interventions of each session were. (now added 189 ff.) The facial expressions were assessed without content of what was said, by muting the videos. The transcripts were conducted thereafter. ( now added 161 ff.)  

RESULTS

the first table should be a summary of all the scores on FACS, PIL and WAI along with patient and therapist characteristics Wasn't the type of intervention in table 1 "interpretive" and not "confrontive" perhaps you need to show the combined "all therapy types" in another row? where are the other facial affects (anger, sadness, fear) in the table?

The difference in WAI scores between types of therapists seemed striking and perhaps much more significant.  it would be good to comment on this impact of skill level on the type of therapy and working alliance. 

I would be curious about how these findings would change if one were to only look at the FACS of only the experienced therapists. 

The other facial affects (anger, sadness, fear) were left out of the table as there were so few of them. (now added row 237 ff.) There was no significant correlation between the clinicians experience and other measures. However, we found that sessions with psychoanalytically trained clinicians scored a higher WAI. We reported and added this (2545ff.).

Many thanks.

Reviewer 2 Report

As a Counselor Educator, this was an interesting article to read.  The implications for counselor education are vast.  The author's pulled together a variety of threads to examine an intricate issue.  Due to the complexity, the article got a little 'weedy' in the results section.  The average reader even with high interest would have to really concentrate on the different implications and interrelated factors.  A relationship graph if possible would have been helpful. 

From a practical application stand point, it is exciting.  Not only did the article examine fine details of therapeutic relationship, but the importance of detailed theory specific focus in counselor education, and a personal favorite, the findings that some of the more perceived 'unpleasant' experiences in the therapeutic relationship could be necessary elements for quality.  

Well done. Thank you for your work to advance our field. 

Author Response

Many thanks for the supporting statements!

Reviewer 3 Report

I found this paper interesting and innovative, even if with a reduced sample.

I have only some minor concerns

Introduction

You can insert this sentence at line 53 page 2. “Moreover, there is a paucity of studies on the correlation between nonverbal and verbal behavior in therapeutic treatment (Henry, 2012; Bänninger-Huber & Peham, 2009), even if in the chronic illness such as cancer they are taken into consideration and studied as communication strategies to improve non-verbal behaviors (Tremolada et al., 2011).

For your reference: Tremolada M., Bonichini S., Pillon M., Schiavo S., Carli M. (2011). Eliciting adaptive emotions in conversations with parents of leukemic children receiving therapy. Journal of Psychosocial Oncology, 29(3): 327-346.

Methods

At line 106: You intend “adult age”? Insert adult in the sentence.

At what time of the therapy the interviews took place?

Why you didn’t assess possible patients’ gender differences? Are female patients more expressive? There are gender differences along non-verbal behaviors? And what about therapists?

These considerations could be added as recommendations for future research.

Author Response

Introduction

You can insert this sentence at line 53 page 2. “Moreover, there is a paucity of studies on the correlation between nonverbal and verbal behavior in therapeutic treatment (Henry, 2012; Bänninger-Huber & Peham, 2009), even if in the chronic illness such as cancer they are taken into consideration and studied as communication strategies to improve non-verbal behaviors (Tremolada et al., 2011).

For your reference: Tremolada M., Bonichini S., Pillon M., Schiavo S., Carli M. (2011). Eliciting adaptive emotions in conversations with parents of leukemic children receiving therapy. Journal of Psychosocial Oncology, 29(3): 327-346.

This has been inserted into the article as suggested. Thank you for the suggestion.

Methods

At line 106: You intend “adult age”? Insert adult in the sentence.

Corrected.

At what time of the therapy the interviews took place?

Patients were at varying stages of their respective therapeutic interventions ; we did not target a specific time-point in treatment in our study. This may be an interesting point of investigation for future research.

Why you didn’t assess possible patients’ gender differences? Are female patients more expressive? There are gender differences along non-verbal behaviors? And what about therapists?

Actually, we did carry out analyses on the effect of gender with respect to the frequency of expression of basic emotions, both in patients and in therapists alike, but had originally elected not to include these data in the paper, as no statistically significant differences were found by our analysis.

However, we have now added a couple of short sentences (lines 219-223) to help make that clear:

No significant differences in terms of the frequency of expression of any of the basic emotions were found between male and female therapists or male and female patients, respectively. Also, whether patients or therapists were paired with someone of the same gender or not in the interviews similarly had no significant impact on such frequency of expression.

Round 2

Reviewer 1 Report

the first table should be a summary of all the scores on FACS, PIL and WAI along with patient and therapist characteristics

Author Response

Dear reviewer, thank you for sticking to this suggestion, we now incorporated the table giving an overview of the FACS, PIL, WAI descriptive statistics (Patient/Therapist), we hope to have created an interesting manuscript for the readers of the journal, kind regards, Henriette Löffler-Stastka 
